# Divergent *Plasmodium* actin residues are essential for filament localization, mosquito salivary gland invasion and malaria transmission

**Michelle Yee**[1¤a], **Tobias Walther**[1¤b], **Friedrich Frischknecht**[1,2]*, **Ross G. Douglas**[1,3]*

**1** Integrative Parasitology, Center for Infectious Diseases, Heidelberg University Medical School, Heidelberg, Germany, **2** German Centre for Infection Research, DZIF, partner site Heidelberg, Heidelberg, Germany, **3** Biochemistry and Molecular Biology, Interdisciplinary Research Centre and Molecular Infection Biology, Biomedical Research Centre Seltersberg, Justus Liebig University Giessen, Giessen, Germany

¤a Current address: Max Planck Institute for Infection Biology, Berlin, Germany
¤b Current address: Max Planck Institute for Medical Research, Heidelberg, Germany
* freddy.frischknecht@med.uni-heidelberg.de (FF); ross.g.douglas@ernaehrung.uni-giessen.de (RGD)

**Data Availability Statement:** All relevant data are within the manuscript and its Supporting Information files.

## Abstract

Actin is one of the most conserved and ubiquitous proteins in eukaryotes. Its sequence has been highly conserved for its monomers to self-assemble into filaments that mediate essential cell functions such as trafficking, cell shape and motility. The malaria-causing parasite, *Plasmodium*, expresses a highly sequence divergent actin that is critical for its rapid motility at different stages within its mammalian and mosquito hosts. Each of *Plasmodium* actin's four subdomains have divergent regions compared to canonical vertebrate actins. We previously identified subdomains 2 and 3 as providing critical contributions for parasite actin function as these regions could not be replaced by subdomains of vertebrate actins. Here we probed the contributions of individual divergent amino acid residues in these subdomains on parasite motility and progression. Non-lethal changes in these subdomains did not affect parasite development in the mammalian host but strongly affected progression through the mosquito with striking differences in transmission to and through the insect. Live visualization of actin filaments showed that divergent amino acid residues in subdomains 2 and 4 enhanced localization associated with filaments, while those in subdomain 3 negatively affected actin filaments. This suggests that finely tuned actin dynamics are essential for efficient organ entry in the mosquito vector affecting malaria transmission. This work provides residue level insight on the fundamental requirements of actin in highly motile cells.

## Author summary

Actin is one of the most abundant and conserved proteins known. Actin monomers can join together to form long filaments. The malaria-causing parasite is transmitted by mosquitoes and needs actin to move very rapidly. An actin from the parasite is different to other actins: its amino acid sequence has relatively high amounts of changes compared to

**Funding:** This work was funded by the Human Frontiers Science Program (HFSP, grant number RGY0066/2015) and the Deutsche Forschungsgemeinschaft (DFG, German Research Foundation) – Projektnummer 240245660 - SFB 1129 (http://www.sfb1129.de) (to FF). RGD is funded by the LOEWE Centre DRUID ("Novel Drugs Targets against Poverty-related and Neglected Tropical Infectious Diseases") within the Hessian Excellence Program. TW was supported by the Studienstiftung des deutschen Volkes e.V.. The funders had no role in study design, data collection and analysis, decision to publish, or preparation of the manuscript.

**Competing interests:** The authors have declared that no competing interests exist.

animal species and the actin tends to form only short filaments. We previously identified two large parts of the protein that were critical for the parasite since these large parts could not be exchanged with the equivalent regions of other species. In this study, we focused in on these regions by making more discrete mutations. Most mutations of the actin sequence were tolerated by the parasite in the blood stages. However, these mutants has striking defects in progressing through mosquitoes, especially in invading its salivary glands. We used a new filament labeler to visualize how these mutations affect the actin filaments and found surprisingly different effects. Taken together, small changes to the sequence can have large consequences for the parasite, which ultimately affects its ability to transmit to a new host.

## Introduction

Actin is a highly conserved and ubiquitous polymer-forming eukaryotic protein [1]. The actin monomer consists of four subdomains that make up a central nucleotide-binding cleft, which houses ATP or its hydrolysis products [2,3]. Actin's ability to typically form dynamic yet stable filaments makes it a central player in various cellular functions including trafficking, cell shape maintenance and substrate-based motility. Various modes of motility have been described. Mesenchymal and amoeboid modes of motility involved striking extensions of plasma membranes to facilitate their movement, while gliding motility involves no apparent changes in cell shape [4–8]. While these modes of substrate-based motility are observably different and depend on cell type, molecular signals, and the surrounding environment, a unifying feature between these modes is the essential role of dynamic turnover of actin filaments to mediate cell locomotion.

The causative agent of malaria, the *Plasmodium* parasite, employs gliding motility in multiple steps of its life cycle to penetrate different organs of its vertebrate and mosquito hosts [9]. A subset of blood stage forms of the parasite differentiate into sexual forms, the gametocytes, which are transmission competent and capable of further development in mosquitoes. After a mosquito blood meal, activated male and female gametes fertilize and develop into ookinetes, which subsequently employ gliding motility to penetrate the mosquito midgut epithelium [10,11]. Successfully traversed ookinetes transform into oocysts, which over the course of approximately two weeks produce hundreds of sporozoites per oocyst. These sporozoites egress from oocysts [12] and are passively transported by the mosquito circulation (the hemolymph). Upon encountering the mosquito salivary glands, the sporozoites actively penetrate this organ. When the mosquito then probes for a blood meal, sporozoites are deposited in the skin of a new host and rapidly move by gliding motility in the skin until contacting blood capillaries in which they traverse and are once more passively transported with blood flow until reaching the liver [13,14]. Sporozoites once more engage in active gliding to traverse Kupffer or endothelial cells and several hepatocytes before developing in a final host hepatocyte [15,16]. After rapid asexual development, exoerythrocytic forms are released into the bloodstream and invade red blood cells thus completing the parasite's complex life cycle [17,18].

The parasite therefore requires gliding motility for transmission to, through and from mosquitoes. While ookinetes move at speeds of 0.05 to 0.2 μm/s (similar to migrating leukocytes) and can glide for at least 24 hours, sporozoites move for up to about 1 hour at 10 times faster speeds [19]. Efficient gliding is mediated by the dynamic interplay between myosin class XIV motors, actin filaments and associated membrane spanning adhesins [8, 20], with distinct self-

organised actin filaments likely accounting for directional motility [21]. *Plasmodium* expresses two actin isotypes. Actin 2 contributes to the maturation of male gametes and oocyst formation, while actin 1 is expressed throughout the life cycle and is the primary contributor to cell invasion and motility [22–25]. The *Plasmodium* actin cytoskeleton displays unusual modifications from canonical opisthokont systems. Firstly, *Plasmodium* appears to express a highly reduced repertoire of conserved predicted actin binding proteins to regulate monomer-filament dynamics [26]. Secondly, actin 1 is biochemically different: it has a modified filament architecture, is unable to be stained with phalloidin, is resistant to latrunculin, displays increased filament disassembly rates and, while it has the ability to form transiently long filaments, ultimately forms very short filaments of approximately 100 nm in length [27–34]. Understanding the contributors of these differences in *Plasmodium* filament dynamics provides important insights into the fundamental properties of actin filaments and the requirements of such dynamics in highly rapid motility across eukaryotes.

The tertiary structure of *Plasmodium* actin 1 is very similar to canonical actins [31], yet *Plasmodium* actins are among the most divergent eukaryotic actins known with a sequence identity less than 80% compared to higher eukaryotes. Changing the sequence composition to opisthokont equivalents has consequences at the biochemical and cellular level for apicomplexans [33,35,36]. We previously investigated the contribution of each subdomain towards divergent actin dynamics and the consequences of these exchanging subdomains to vertebrate equivalents on parasite progression in mosquito and mammal. This approach identified subdomains 1 and 4 as critical contributors for rapid and smooth motility in the mosquito stages, while subdomains 2 and 3 were unable to be exchanged revealing that their divergent nature is important for parasite viability in the mammal, where transfection and selection of mutants is performed. Further, changes in subdomain 2 and 3 improved incorporation of parasite actin into mammalian filament networks, suggesting that divergent amino acid residues in these regions are essential for proper monomer-filament interactions [36]. Subdomains 2 and 3 have been reported to provide important interactions for longitudinal and lateral contacts between monomers in the filament as well as for important binding sites for various actin binding proteins [33,37,38]. However, the contributions of the divergent amino acid residues in these subdomains to *Plasmodium* actin 1 function remains unclear since the lethal phenotypes brought about by subdomain exchange do not allow for further study. In this paper, we explore the contribution of discrete single and multiple amino acid residue changes of subdomains 2 and 3 and their contribution to altered *Plasmodium* actin function. We also employed an actin chromobody (ChromoTek) [39–41] to visualize the alterations to actin filament localization brought about by mutation. This showed that motile mosquito stages are strongly affected by mutations in these subdomains in motility and organ penetration. We also present the first visualization of actin filaments in *Plasmodium* sporozoites in both wild-type and mutated lines, revealing that different mutations have surprisingly different consequences, with only a single change in primary sequence affecting localization and possibly rendering filaments more stable.

## Results

### Divergent amino acid residue exchanges in subdomains 2 and 3 yield viable parasites

Subdomains 2 and 3 provide important interactions both within the filament and with actin binding proteins. Interestingly, while some regions of *Plasmodium* actin 1 are relatively well conserved (such as subdomain 1), regions in subdomains 2 and 3 contain some of the most divergent stretches of sequence when compared to canonical actins (**Fig 1A–1C**). We selected

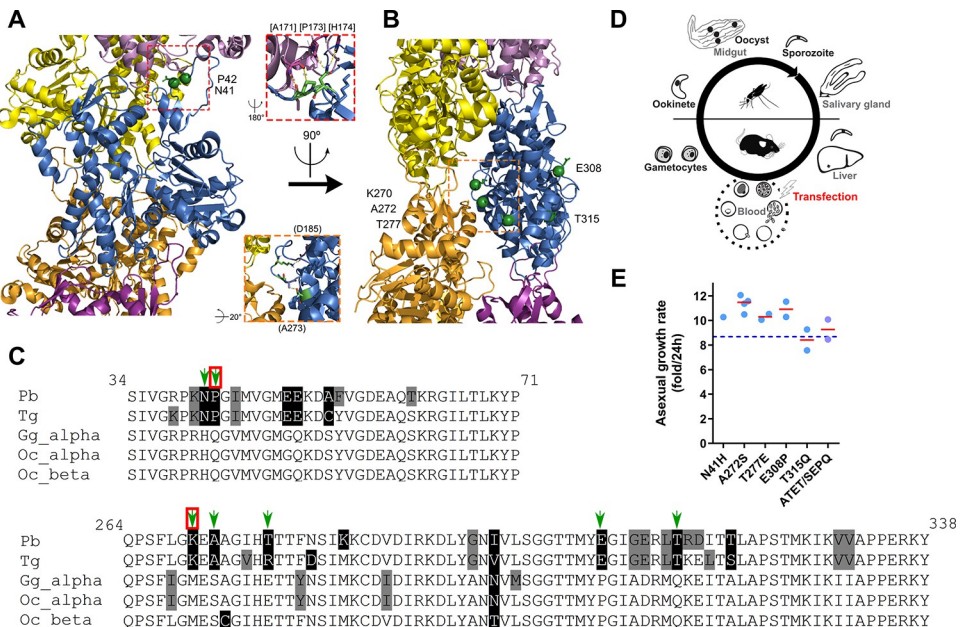

**Fig 1. *Plasmodium* actin 1 has divergent amino acid residues in subdomains 2 and 3.** (A) and (B) *Plasmodium* actin monomer within the filament (PDB 5OGW, *P. falciparum*) [38]). The central monomer is shown in blue with neighbouring monomers in various other colours. Residues of interest are indicated as green balls. Red and orange dashed inset boxes show close-ups of relevant regions and marked residues in subdomain 2 and 3 respectively within the filament. Residues with round brackets indicate nearby intra-molecular neighbours and residues with square brackets indicate nearby inter-molecular neighbours. Thus, changes in residues of interest could have consequences for actin monomer intra- and inter-molecular contacts and/or packing. Images generated with open source PyMOL. (C) Relevant sequence sections of *Plasmodium* actin subdomains 2 and 3 reveal highly sequence divergent positions between apicomplexans and vertebrates. Green arrows indicate the residues of interest, red boxes indicate residues that we previously attempted unsuccessfully to mutate in the parasite [36]. Pb: *Plasmodium berghei* Tg: *Toxoplasma gondii*, Gg_alpha: *Gallus gallus* alpha skeletal muscle actin, Oc_alpha: *Oryctolagus cuniculus* aortic smooth muscle actin, Oc_beta: *Oryctolagus cuniculus* beta actin. (D) Schematic representation of the *Plasmodium* life cycle showing relevant stages. Transfection and selection of genetic modifications are performed with asexual blood stages. (E) Single asexual parasites were i.v. injected into naïve mice and growth rates determined. All mutants grew at rates comparable to wild-type (dashed line indicating growth rate as published previously, [36]). Red line indicates median value. Note that only one clone of mutant N41H could be obtained.

particular amino acid residues for more detailed study based on three criteria: 1) the *Plasmodium* actin residue had a striking change in chemical properties compared to their corresponding vertebrate counterpart; 2) the corresponding vertebrate counterpart is conserved across vertebrates and isotypes suggesting critical conservation; and 3) the divergent residues are present in key contact sites either within filaments or potentially towards actin binding proteins. Using these criteria, we earmarked seven divergent residues for mutation: N41H, P42Q (both found on the D-loop of subdomain 2), K270M, A272S, T277E (found on or near the H-plug of subdomain 3), E308P and T315Q (found on the exterior on subdomain 3). While some of these residues do not have direct bonding to neighbouring monomer residues (see inset close-ups), they could mediate intramolecular interactions and/or mutation of these residues to vertebrate counterparts could provide alternative contacts, either by modifying loop dynamics or the vertebrate equivalent forming contacts that do not occur in the native *Plasmodium* filament, and thus alter filament dynamics. We employed our two-step selection approach [36] to transfect parasites in the blood stages of the life cycle of *P. berghei* and assess if or where phenotypic consequences occur (**Fig 1D**). We previously attempted the mutations P42Q and K270M in *P. berghei* but could not replace the wild-type gene, indicating that these mutants

confer important interactions that are essential for parasite blood stage development [36]. For the remaining individual mutations in subdomains 2 and 3, as well as a quadruple subdomain 3 mutant (a combined mutant of A272S, T277E, E308P and T315Q; termed ATET/SEPQ), we were able to obtain transfectants possessing the desired residue change(s) (**S1 Fig**). Remarkably, these demonstrated no obvious defects in blood stage asexual growth demonstrating that any alteration of actin dynamics mediated by these changes does not manifest at this stage (**Fig 1E**).

## Combined mutations in subdomain 3 impaired parasite migration in early mosquito stages

We next assessed the ability of these mutants to infect the mosquito host (**Fig 2A**). *Anopheles* mosquitoes were allowed to feed on infected anesthetised mice and total oocysts per infected mosquito determined. With the exception of A272S, which showed slightly reduced oocyst levels, individual mutants displayed similar oocyst loads to the wild-type control experiment and thus proceeded normally through the initial stages of mosquito infection (**Fig 2B**). However,

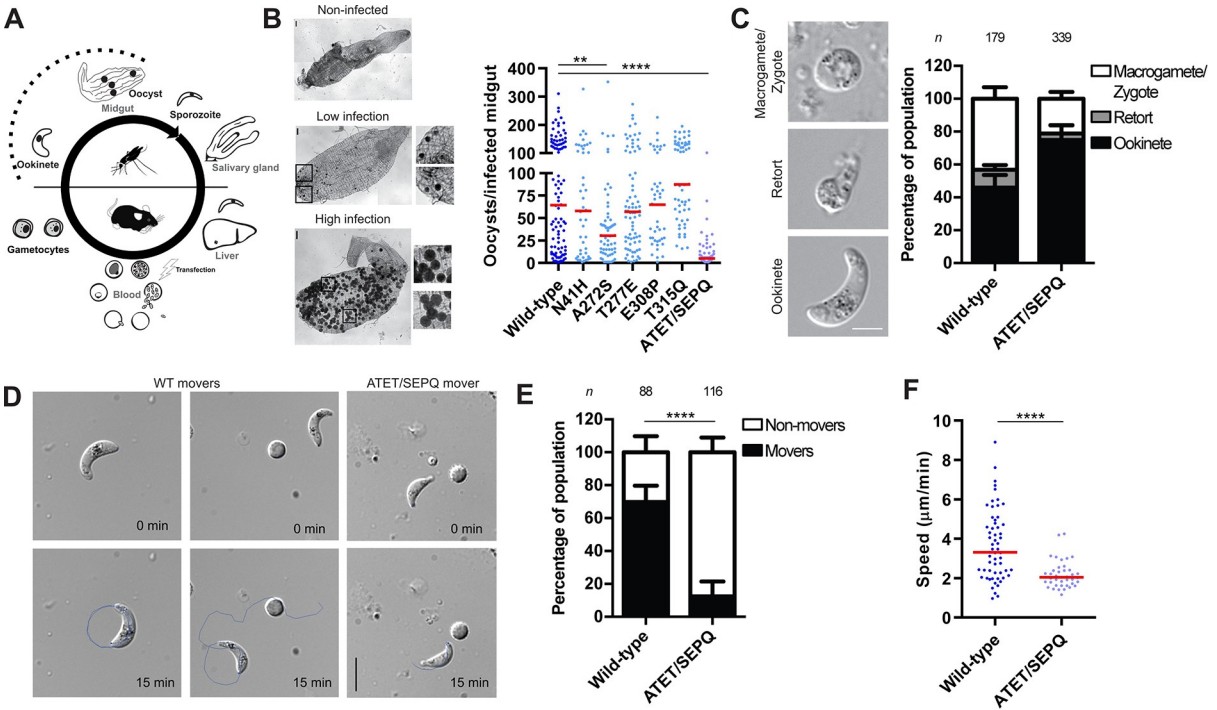

**Fig 2. Single mutations of subdomain 2 and 3 have no effect on early mosquito stage infection while a quadruple mutant of subdomain 3 has highly reduced ookinete motility and infectivity.** (A) A schematic representation of the life cycle with early mosquito stages indicated with a dashed line. Ookinetes are motile parasites that develop in the mosquito midgut, penetrate the midgut epithelium and subsequently develop into oocysts. (B) Representative images of different oocyst loads observed in mosquito midguts and oocyst loads for different parasite lines. All mutants showed normal ranges of infections with the exception of quadruple subdomain 3 mutant ATET/SEPQ, which was dramatically reduced compared to the control. T315Q had slightly increased oocyst levels (p = 0.03). Red line indicates median value. Mann-Whitney test, **p<0.01; ****p<0.0001, $n \geq 44$ infected midguts from at least 2 mosquito infection experiments. Scale bar: 100 μm. Midgut images were stitched to form composite image of whole organ. (C) Representative images of ookinete development. After gamete fusion, the round zygote develops through an intermediate retort stage into a mature ookinete. Scale bar: 5μm. Quantification of ookinete developmental stages after 21 hours post activation reveals no defect in ookinete development for the ATET/SEPQ mutant. Bars represented as mean ± standard deviation. (D) Representative stills at acquisition start (0 min) and end (15 min) for motile wild-type (WT) and ATET/SEPQ ookinetes. Scale bar: 10 μm. (E) ATET/SEPQ has a significantly reduced population of motile ookinetes. Bars represented as mean ± standard deviation. Fischer's exact test, **** p<0.0001. (F) Quantification of speeds for motile parasites indicated that the motile population of the ATET/SEPQ mutant line moved slower than wild-type. Red line indicates median value. Mann-Whitney test, **** p<0.0001.

the ATET/SEPQ mutant showed highly reduced total oocyst numbers (**Fig 2B**), implying that the parasite is defective early on in transmission. The parasite undergoes rapid and highly controlled developmental steps upon transmission to the mosquito vector. Upon fertilization, the round zygote becomes polarized and develops into an intermediate form (termed retort), which subsequently completes transformation into a mature ookinete within approximately 21 hours (**Fig 2C**). ATET/SEPQ developed into mature ookinetes in a manner suggesting no defects with this mutant in early mosquito infection (**Fig 2C**). While development remained unaffected, ATET/SEPQ had highly reduced numbers of moving ookinetes (**Fig 2D and 2E**) and this motile population showed a further decrease in parasite speeds compared to wild-type controls (**Fig 2F**). Thus, a decrease in both motile ookinetes and their speed accounts for the highly reduced mosquito infectivity of this mutant.

## Single actin mutant sporozoites were impaired in salivary gland invasion, yet only the subdomain 2 single mutant exhibited aberrant motility

Hundreds of sporozoites egress from a mature oocyst into the hemolymph and ultimately invade the mosquito salivary gland (**Fig 3A**). We next investigated the effects of actin mutation on the motility and invasion capacity of sporozoites. We first analyzed hemolymph-derived sporozoites. Individual subdomain 3 mutants had similar proportions of motile populations and speeds as that of the wild-type control (**Fig 3B and 3C**). ATET/SEPQ had a drastic reduction in its motile population, similar to the observation of this mutant at the ookinete stage, and only 2 motile hemolymph sporozoites were observed out of a total of 430 parasites. This motility defect could not be rescued by adding small amounts of actin stabilizing agent Jasplakinolide previously reported to influence retrograde flow and force transduction and rescue motility defects of a genetically modified sporozoite [25,42,43] (**S2 Fig**), hinting that the defect observed might be due to loss of actin binding protein interaction and not simply changes in monomer contacts within the filament (see below). Interestingly, subdomain 2 mutant N41H also had a large reduction in its motile sporozoite population and the few parasites classed as movers moved significantly slower than the wild-type indicating that this mutation has pronounced effects on actin dynamics at this stage (**Fig 3B and 3C**). Comparison of attaching versus floating sporozoites revealed similar proportions across lines, suggesting that none of these mutations affect adhesion in our 2D gliding assay (**Fig 3B**). All mutants had an approximately 10-fold reduction in salivary gland sporozoite numbers, indicating that all lines struggle to invade the insect salivary glands (**Fig 3D**). It is worth noting that even the individual subdomain 3 mutants, which displayed no apparent defects in hemolymph-derived sporozoite motility, also had salivary gland invasion defects to the same degree as that of motility deficient lines. In terms of salivary gland sporozoites, very few ATET/SEPQ sporozoites were observed since there is a cumulative loss of parasites as they progressed through the mosquito. In our assay, out of the 64 observed ATET/SEPQ parasites, we observed neither movers nor partial movers yet the proportion of non-movers was similar to the proportion of attached sporozoites in the wild-type control (classed as "non-movers", "partial movers" and "movers") indicating once again that adhesion remains unaffected (**Fig 3E**). E308P had a decreased motile population with E308P and T315Q displaying slightly reduced speeds, while the other subdomain 3 mutations displayed motility patterns and speeds very similar to the wild-type (**Fig 3E–3G**). While the subdomain 3 mutants generally showed a largely consistent pattern between hemolymph and salivary gland sporozoites, N41H was variable: salivary gland sporozoites had a similar proportion of motile parasites but showed a slight decrease in average speed, thus having a less striking phenotype once these sporozoites have invaded this organ.

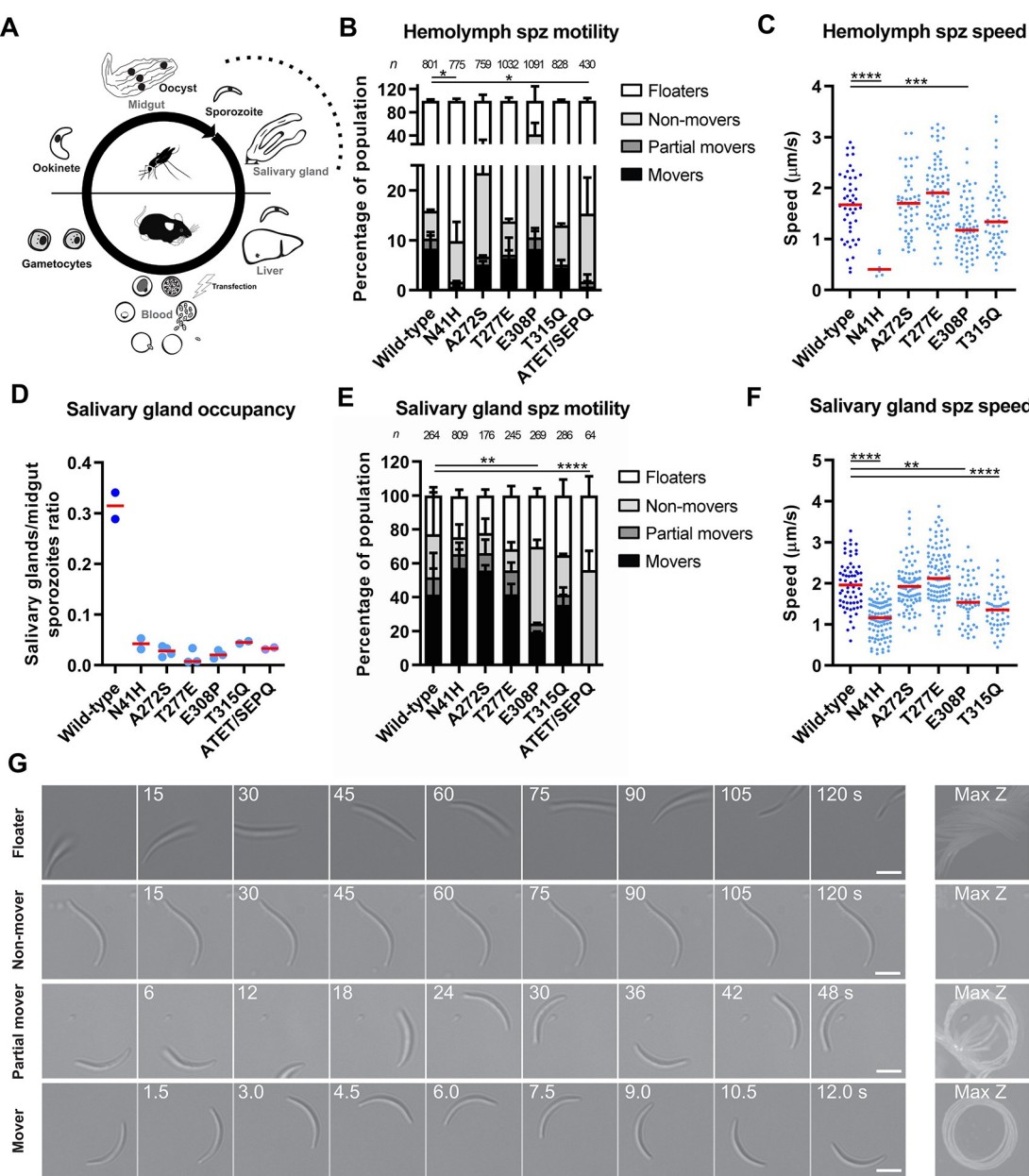

**Fig 3. Actin mutants are highly deficient in salivary gland invasion.** (A) Schematic representation of the *Plasmodium* life cycle with late mosquito stages indicated with a dashed line. Sporozoites egress from oocysts and, after passive transport through the mosquito hemolymph, penetrate salivary glands. (B) Hemolymph sporozoite motility was analysed using a 2D motility assay. Consistent with the effect of ATET/SEPQ on ookinete motility, this mutant was highly deficient in motility. Subdomain 2 mutant N41H also displayed a highly reduced motile population while single subdomain 3 mutants were motile at similar level to wild-type controls. Bars represented as mean ± standard deviation for repeats between at least two independent mosquito infections (with exception of N41H, standard deviation of two technical repeats across two days from single infection). Fischer's exact test, *p<0.05. (C) Speeds of moving hemolymph sporozoites were similar between mutants, with the exception of N41H which was slower. E308P was also slower but was in the normal median range for wild-type speeds. ATET/SEPQ is absent since only two motile sporozoites were observed in this assay. Red line indicates median value. Mann-Whitney test, *** p = 0.0002, ****p<0.0001. (D) All actin mutants displayed an approximately 10-fold reduction of salivary gland sporozoite numbers when normalized to midgut sporozoite loads of the same mosquitoes. Red line indicates median value of repeats of d19 counts of least two independent mosquito infections per line. (E) Single mutant salivary gland sporozoites generally had a similar proportion of motile parasites while E308P had a slight reduction and ATET/SEPQ had zero movers. Bars represented as mean ± standard deviation for repeats between at least two independent mosquito infections. Fischer's exact test, **p<0.01, ****p<0.0001. N41H displayed a slightly elevated motile population (p = 0.03) but this is in the normal range for wild-type parasites. (F) Subdomain 3 mutant salivary gland sporozoites had similar speeds when compared to the wild-type control. N41H, E308P and T315Q showed

slight reductions in speed. ATET/SEPQ is absent since no motile salivary gland sporozoites were observed in this assay. Red line indicates median value. Mann-Whitney test, **p<0.01, ****p<0.0001. T277E had a slightly increased speed (p = 0.02) likely due to experimental variation. (G) Example images of different categories used to class motility in (B) and (E). Number indicates time in seconds; scale bar: 5 μm. For examples of other patterns classed as "non-movers", see **S3 Fig**.

### Actin mutants are less transmissible due to deficiency in salivary gland invasion

We next assessed whether actin mutations affected infection of a new mammalian host (**Fig 4A**). In order to investigate whether hemolymph-derived sporozoites were generally less infectious, we injected 10 000 hemolymph sporozoites and monitored their appearance and growth in the blood stage. The ATET/SEPQ mutant was very poorly infectious, with only one mouse developing a detectable blood stage parasitemia and this patency was considerably late (**Fig 4B**). Interestingly, all other mutants displayed very similar parasite progression post-injection indicating that there are no detectable defects in hepatocyte invasion, development or egress. Importantly, this further demonstrates that these mutants are not generally "invasion incompetent", but appear to be specifically defective in salivary gland invasion. Sporozoites deposited in the skin by natural transmission (i.e. by mosquito bite) had generally lower parasitemias at day 6 post-infection and infected fewer mice (**Fig 4C**). Given that salivary gland sporozoites loads are highly reduced for the mutant lines, it is likely that lower infection rates are simply due to fewer sporozoites deposited per bite. Further, intravenous injection of salivary gland sporozoites resulted in highly similar infection rates across lines suggesting that overall infectivity is not reduced (**Fig 4D**) but that the decrease in transmission is due to decreased invasion efficiency of sporozoites into the salivary glands.

### The actin chromobody visualizes *in vivo* sporozoite actin localization

Mutations in subdomains 2 and 3 strongly affected parasite infectivity, especially in salivary gland invasion. We sought to visualize actin in the context of this striking phenotype. Previous attempts to visualize actin filaments have been hampered by the highly divergent nature of *Plasmodium* actin and the very rapid motility of sporozoites. Recently, an actin chromobody has been employed to label actin filament structures in both *Toxoplasma* tachyzoites and *P. falciparum* blood stages [44–46]. In order to assess the effectiveness of this probe in sporozoites, we initially transfected wild-type parasites with a chromobody-emerald construct under the control of the actin 1 promoter (**S4 Fig**). Introduction of the chromobody had minimal effects on hemolymph and salivary gland sporozoite motility patterns with minor effects on speeds and, interestingly, resulted in a moderate decrease of salivary gland resident sporozoites (**S5 Fig**). Yet, the line provided plentiful numbers of salivary gland sporozoites for their quantitative characterization. We observed different sub-cellular localizations of the chromobody in the adherent sporozoite populations (**Fig 5A and 5B**). Treatment with the actin filament disrupting compound Cytochalasin D (CytoD) resulted in a uniform whole cell distribution. Treatment with filament stabilizing compound Jasplakinolide (Jas) resulted in a distinct collection of chromobody signal at the front and back of the sporozoite (**Fig 5A and 5B**), reminiscent of responses to Jas with fluorescently tagged actin that was expressed as an additional copy [36,47]. Categorization and quantification of localizations revealed that hemolymph and salivary gland sporozoites have similar distributions and were equally responsive to CytoD (whole cell localization) or Jas treatments (front and back localization) (**Fig 5C and 5D**). The majority of localizations in untreated sporozoites were found to be associated with the back of the sporozoite. A signal only at the back of the sporozoite was the highest individual category observed, with "nucleus and back" and "front and back" localizations also featuring noticeably

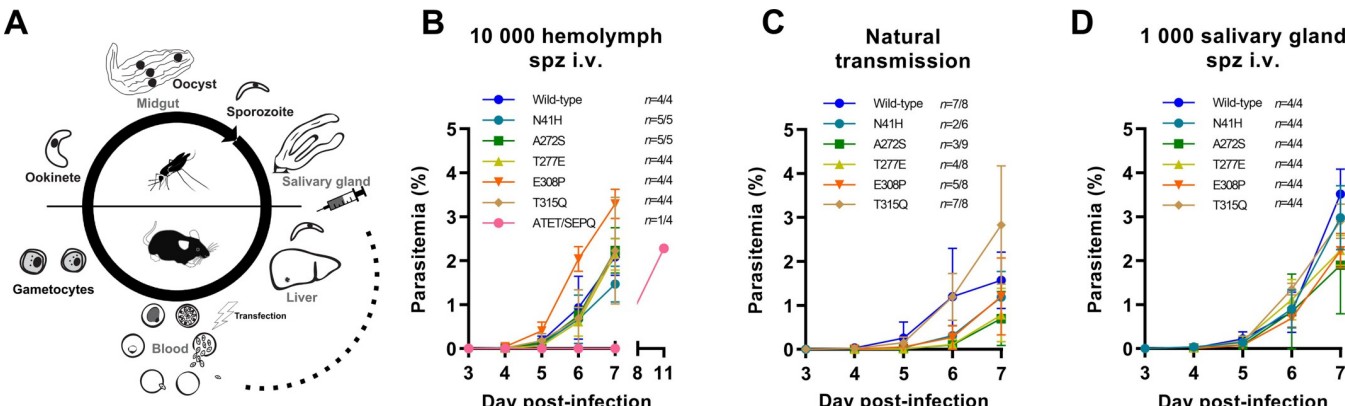

**Fig 4. Actin mutants are infectious for new mammalian hosts yet decreased in natural transmission.** (A) Schematic representation of the *Plasmodium* life cycle with relevant stages indicated with a dashed line. Sporozoites are deposited in the skin of a new mammalian host during mosquito probing for a blood vessel. Sporozoites exit the skin through entry into blood capillaries and are transported to the liver where they invade hepatocytes. Sporozoites transform into exoerythrocytic forms within the host cell which releases merozoites into the bloodstream within approximately three days post-infection. (B) Intravenous injection of hemolymph sporozoites indicated that single mutants (which were deficient in salivary gland invasion) infected mice with similar kinetics to the wild-type control and therefore have no major deficiencies in liver invasion or development. ATET/SEPQ was poorly infectious, with only one mouse out of four becoming patent and with a substantial delay. (C) Natural transmission experiments through mosquito bites shows delays and decrease in infectivity testifying to the essentiality of the divergent amino acid residues. This decrease is best interpreted as due to the remarkably reduced loads of salivary gland sporozoites deposited in the skin. (D) Intravenous injection of salivary gland sporozoites results in infection rates and blood stage progression that are very similar to wild-type levels. Data points represented as mean ± standard deviation. *n* indicates the number of mice positive in relation to the number of mice injected or bitten.

albeit at lower percentages (**Fig 5C and 5D**). Interestingly, these sub-cellular localizations appeared to remain constant during motility (**Fig 5E**) and only infrequently did we observed a change in signal distribution (**S6 Fig**). Taken together, the actin chromobody is a robust tool that specifically labels actin filament localizations, displays similar localizations between sporozoite phases and can detect shifts in actin monomer-filament modulations.

### Actin mutations have different consequences on hemolymph sporozoite actin localization

The chromobody was then employed to determine what effects the mutations had on sporozoite actin localization. We focussed on hemolymph-derived sporozoites since we were interested in the actin localization of the entire population before salivary gland invasion and the study of hemolymph sporozoites allowed for a more quantitative assessment of alterations. One mutant from each subdomain displaying the most extreme phenotype was selected for further analysis. All lines responded to Jas treatment indicating that the actin mutants are functional for polymerization (**Fig 6A-F**). The chromobody also allowed for a coarse-grained estimation of expression levels, revealing that mutants generally did not have major changes in their expression levels, with the exception of N41H, which might have slightly reduced functional protein levels (**S7 Fig**). Remarkably, subdomain 2 mutant N41H showed, in an untreated state, a localization of "front and back" that resembled Jas treated wild-type cells (**Fig 6A and 6B**). While treatment of the mutant with Jas did not strikingly increase the number of sporozoites with "front and back" localization, treatment with CytoD abrogated the localization, just as it did in the wild-type. This suggests that this single mutation potentially increases the filament pool in the sporozoite and might render the parasite actin filaments more stable. Intriguingly, the ATET/SEPQ mutant showed a localization that resembled that of CytoD treated wild-type cells, indicating that these combined mutations probably have a negative effect on filament formation or actin binding protein interaction (**Fig 6C and 6D**).

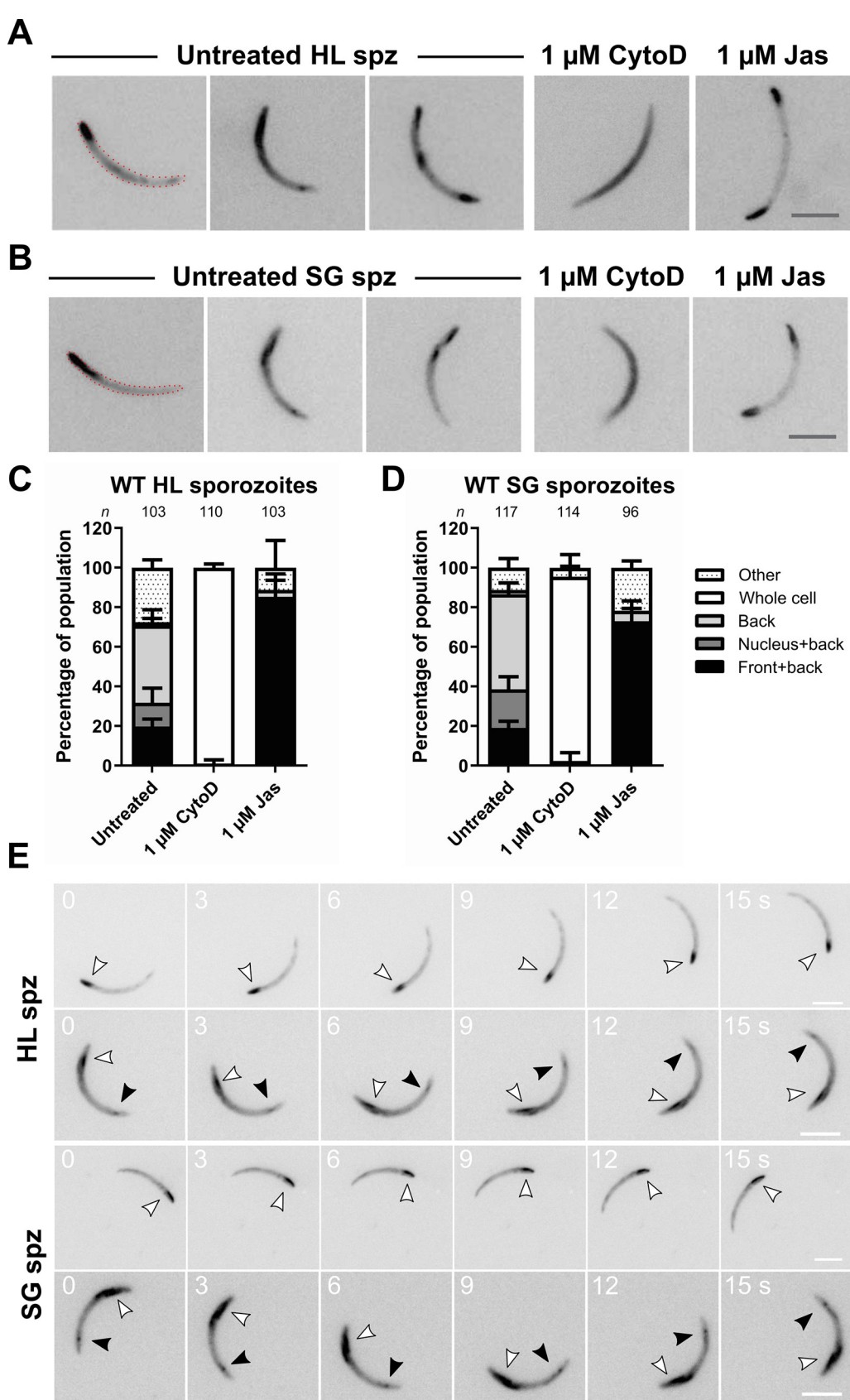

**Fig 5. The actin chromobody localises in distinct regions of the sporozoite and is sensitive to chemical modulations.** (A) Representative images of wild-type hemolymph sporozoites (HL spz) under untreated or treated conditions. Treatment with modulator cytochalasin D (CytoD) resulted in a whole cell distribution while jasplakinolide (Jas) resulted in a "front and back" localisation. Scale bar: 5µm (B) Representative images of wild-type salivary gland sporozoites (SG spz) under untreated or treated conditions. Treatment with modulators CytoD and Jas resulted in similar distributions as observed in treated hemolymph sporozoites. Dotted lines on the left images of panels A and B serve to illustrate the approximate outline of the parasite. (C) and (D) Quantifications of chromobody localisation in the motile and attached population. Localisations associated with the back were the predominant populations observed. Treatment with modulators resulted in a nearly exclusive localisation. Bars represented as mean ± standard deviation across independent assays. (E) Time lapses of motile chromobody expressing sporozoites. Localizations of chromobody signal typically remained constant during imaging. Black and white arrowheads indicate the signals at the front and back of the sporozoite respectively. Number indicates time in seconds; scale bar: 5 µm.

We previously observed that a mutant line that exchanged actin subdomain 4 (PbS4Oc) resulted in sporozoites that had an aberrant "stop-go" motility pattern and salivary gland invasion phenotype [36]. We performed the same assay with this mutant and found that, similar to the N41H mutant, this multiple mutant resulted in a predominantly "front and back" localization (**Fig 6E and 6F**). This might reflect a shifted dynamic that favours more filaments and possibly increased filament stability. Taken together, our data shows that different mutations have opposing effects on parasite actin localization yet lead to the same final consequence: a decrease in salivary gland entry in the mosquito vector.

## Discussion

In this study, we show that discrete changes in divergent amino acid residues of *Plasmodium* actin subdomains have large consequences for parasite transmission. We have found that single mutations to its sequence can essentially prevent the parasite efficiently invading the salivary glands of the mosquito. Understanding the consequences of these sequence changes *in vitro* and *in vivo* further provides understanding in the contribution of these residues to filament assembly and disassembly for actins in general. Strikingly, a single change to the sequence noticeably changes the filament localization within the sporozoite.

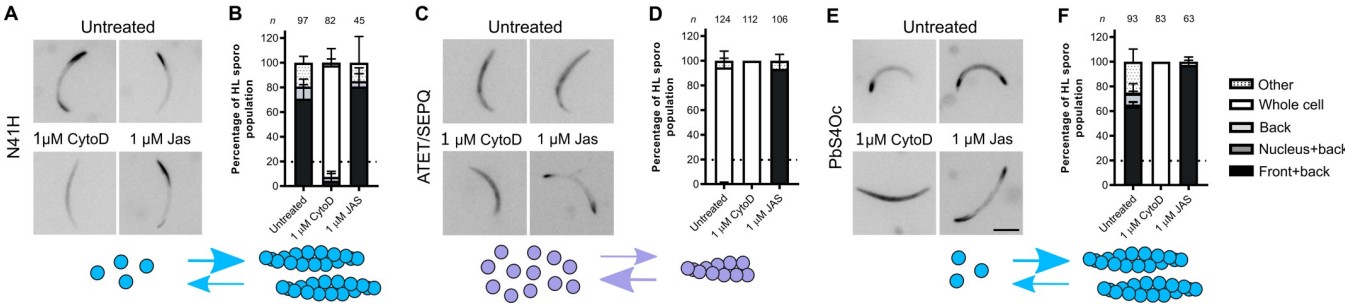

**Fig 6. Actin mutations result in different consequences for hemolymph sporozoite actin localization.** (A) Representative images of chromobody localisation for subdomain 2 mutant N41H untreated and with actin modulators cytochalasin D (CytoD) and jasplakinolide (Jas). (B) Quantifications of localisations in the attached and motile populations. The untreated N41H parasites resemble Jas treated cells indicating a potentially increased filament pool and increased filament stability. (C) Representative images of chromobody localisation for subdomain 3 mutant ATET/SEPQ untreated and with actin modulators. (D) Quantifications of localisations in the attached population (no motile sporozoites were observed since the mutations have a pronounced effect on hemolymph sporozoite motility). The untreated ATET/SEPQ parasites resemble CytoD treated cells indicating a potentially negative effect on the filament pool and filament stability. (E) Representative images of chromobody localisation for subdomain 4 mutant PbS4Oc untreated and with actin modulators CytoD and Jas. (F) Quantifications of localisations in the attached and motile populations. The untreated parasites resemble Jas treated cells indicating a potentially increased filament pool and increased filament stability. Scale bar: 5 µm. Dotted lines indicate relative percentage of "Front + back" localization for the wild-type control as quantified in Fig 5. Bars represented as mean ± standard deviation across independent assays. Cartoons underneath represent possible shifts in actin monomer and filament amounts based on changes of chromobody signal.

## The role of divergent amino acid residues in *Plasmodium* actin dynamics

Actin is highly conserved across eukaryotes, with opisthokont actins displaying more than 95% sequence identity across isotypes and species. There has thus been a strong selective pressure on eukaryotic actin to maintain its sequence for the fundamental purposes of polymerization and force transduction [48]. Any divergence from this largely canonical sequence must therefore represent a need for the cell to employ additional or divergent mechanisms in its cytoskeleton. *Plasmodium* actin 1 is one of the most highly divergent eukaryotic actins known with an accordingly different biochemistry. Increased actin filament disassembly rates produce (over time) short filaments of approximately 100 nm in length. More recent biochemical studies have revealed that *Plasmodium* actin has a similar assembly rate to canonical actins, even to the point of transiently forming long filaments *in vitro*, yet have considerably higher rates of disassembly or fragmentation [32–34], supporting the suggestion that the *Plasmodium* actin filament is intrinsically geared towards disassembly [26].

## *Plasmodium* actin changes are more sensitive in the mosquito, especially in salivary gland invasion

Parasites with mutations in subdomains 2 and 3 developed normally in the mammalian host yet were defective in progression through the mosquito. Salivary gland invasion was particularly sensitive to modifications in actin sequence. Intriguingly, changes in actin localization resulted in the same effect: less colonization of salivary glands. These observations add further evidence to the hypothesis that the salivary gland represents the most formidable barrier for the parasite to complete its life cycle [49–52]. We had previously hypothesized that smooth continuous motility is a requirement for efficient invasion of salivary glands. While this still holds, we have now observed mutant parasite lines (the individual subdomain 3 mutants) in which hemolymph-derived sporozoites appear to have normal motility in our *in vitro* assay, yet nonetheless are defective in salivary gland invasion at a level similar to motility deficient sporozoites (N41H and ATET/SEPQ). This, in contrast to previous interpretations, indicates that 2D motility alone is not sufficient for efficient salivary gland invasion. Indeed, expression of the actin chromobody did not strikingly affect motility patterns of hemolymph or salivary gland sporozoites and had only minor effects on their speeds. Yet, the noticeable decrease in salivary gland invasion rate of the chromobody further points to the observation that actin dynamics are extremely sensitive in this invasion process.

Single mutant sporozoites were deficient specifically in salivary gland entry and not in invasion generally, since infection of naïve mice with hemolymph-derived sporozoites resulted in infection rates very similar to the wild-type control. The intracellular processes that drive organ invasion by *Plasmodium* should thus not be seen as the same for all stages, but rather that each stage requires the optimal dynamic to penetrate a specific organ. Our work indicates that the intracellular molecular interactions need to be the most tightly regulated for salivary gland invasion. In the case of liver invasion, we speculate that tight binding to liver-resident cells, which are highly abundant in heparan sulfate proteoglycans [53] might allow for sufficient time and engagement of the parasite machinery to allow invasion to take place [54]. The putative receptor(s) required for salivary gland entry might be less prevalent, thus requiring all necessary coordination of the intracellular machinery to be optimal.

Other parameters are clearly involved for an optimal "salivary gland invasion dynamic" apart from motility alone, that is, being able to move and at similar speeds to wild-type sporozoites *in vitro*. At a mechanistic level, we hypothesize that modifications to actin affect the optimal coordination of all molecules required such that efficient penetration can occur. Alternatively, alteration of the actin dynamics could affect transmission of sufficient force to

allow penetration. Indeed, studies with *Plasmodium* myosin A mutants have found that affecting the power stroke has pronounced effects on invasion at different stages of the life cycle [52,55]. Methods are available to investigate force transmission in sporozoites, such as optical tweezers and traction force microscopy, which have carefully delineated force dynamics in *Plasmodium* [25,43,56]. Unfortunately, a similar investigation of hemolymph-derived sporozoites is nearly impossible given the highly reduced number of motile sporozoites compared to salivary gland sporozoites. In addition, temperature variation between hosts presents an important consideration for parasite progression and transmission. *Plasmodium* actin dynamics need to be sufficiently optimal for both body temperatures and thus presents a specific challenge for the parasite in managing the necessary dynamics of each stage in its life cycle.

## Chromobody as a tool for *in vivo* visualization of *Plasmodium* actin in motile sporozoites

We have shown that the actin chromobody is an effective tool to study actin localization in the highly motile sporozoite, both in the case of understanding actin localization employed by wild-type parasites and to investigate the effects of actin modulation. These changes in localization may reflect direct alterations on actin dynamics or changes to binding of interacting proteins. However, it is important to note that the actin chromobody only visualizes the localization of actin filaments and thus further work would be needed to define what exactly is driving these changes in localization. Our data comparing a chromobody expressing line to wild type sporozoites suggests that the chromobody does not affect overall sporozoite motility patterns but might have subtle effects on actin dynamics in terms of speeds and salivary gland invasion. However, we believe that its usefulness far outweighs the potential relatively minor effect on actin dynamics.

In wild-type sporozoites, the primary site of collection of actin filaments is associated with the rear of the parasite. This is consistent with the observation of coronin, an actin filament binding protein, which collects at the back of motile salivary gland sporozoites [49]. We also noticed that there was a smaller population (~10–20%) that had discrete signal at the nucleus and back. Recent work in *Toxoplasma* has shown the presence of an F-actin ring around the nucleus during host cell invasion, and has been postulated to ease nuclear passage by stabilizing the tight junction, facilitate pushing the nucleus through the constriction and protect the genetic information from stress induced by constriction [45]. It is possible that the small population observed with nuclear signal could represent sporozoites that recently underwent constriction, either within hemolymph vasculature or during salivary gland invasion. The "front and back" signal is a minority population in wild-type sporozoites yet becomes the predominant population when subdomain 2 or 4 is mutated. This resembles Jas treated sporozoites and once again suggests the idea that the primary sites of filament formation and turnover could be at the front and back of the sporozoite.

The chromobody probably cannot distinguish between *Plasmodium* actin isotypes 1 and 2. Since mutation of actin 1 results in obvious changes of localization, it is clear that the actin chromobody labels at least actin 1. Proteomic studies have identified actin 2 in sporozoites, although the exact concentrations and functions of actin 2 in sporozoites remain to be determined [57,58]. Chromobodies that are isotype specific would be very useful in the identification of isotype specific cellular locations and functions in the parasite. Further, the chromobody will be a valuable tool to study other stages of the *Plasmodium* life cycle.

## The role of a divergent subdomain 2 amino acid residue

Conversion of a *Plasmodium* asparagine to histidine in subdomain 2 (N41H) resulted in a localisation pattern similar to Jas treated sporozoites and thus possibly more stable filaments

in the *Plasmodium berghei* sporozoite. We had previously identified that this change alone was sufficient for improved incorporation of parasite actin monomers into mammalian filaments [36]. This residue is positioned at the base of the D-loop. This is a dynamic structure, in which its flexibility is critical for filament formation [59, 60] and provides both lateral and longitudinal interactions within the filament [36,38,61,62]. The vertebrate conserved histidine residue is involved in mediating important electrostatic interactions with the H-plug of the neighbouring monomer [36,38,62,63]. When this residue is replaced by an asparagine, as in the case of *Plasmodium* actin, this results in a deterioration of electrostatic interactions at this site that likely contributes to altered filament stability [38]. The chemical nature of this position is clearly of functional importance, as an oxidised histidine in mammalian actin was not able to polymerize [64,65] while the histidine to asparagine conversion in human smooth α-actin is associated with thoracic aortic aneurisms and dissections [66]. It is possible that such alterations in this flexible interface could also have further negative consequences on actin binding proteins as has been documented for canonical actins with coronin and myosin, which bind near to the D-loop [67,68], as well as ADF/cofilin, which is strongly affected by the D-loop conformation [59]. Estimation using chromobody response levels suggests that the mutation might result in a minor decrease of functional hemolymph sporozoite *in vivo* actin protein levels, although this could possibly be due to the limitations of the assay and experimental variation. Even if protein levels of this mutant are reduced, what remains highly striking is that N41H actin can still render localizations indicative of more stable (or increased numbers of) filaments than wild-type sporozoites despite potentially lower concentrations. Given the concentration dependent nature of actin polymerization, this further emphasizes that the mutation likely renders a more "polymerization-prone" actin *in vivo*. Our phenotypic characterisations thus emphasize the critical contribution of this single amino acid residue to parasite filaments *in vivo*.

The potential stabilising effect of this mutation had an effect on the motile population of sporozoites, in particular those in the hemolymph. We saw a comparable effect on chromobody localization with the PbS4Oc mutant (which had a "stop-go" motility pattern), suggesting a similar consequence on actin behaviour. Yet, we observed only relatively few (72 out of 420 motile cells) pausing N41H mutant sporozoites. We suggest that there could be different thresholds of actin dynamics that result in different extents of phenotypes. It is possible that changes to subdomain 4 resulted in a change to both the inherent biochemical properties of the actin itself (potentially rendering its filament slightly more stable) and the ability for actin binding proteins to bind effectively to the mutated actin. Indeed, we have previously reported that *Plasmodium* coronin was not able to bind efficiently to PbS4Oc actin [36]. This suggests that the pausing observed for the PbS4Oc is due to a dual effect of reduced actin turnover (mediated by coronin), coupled with a more stable filament. Given these differences, in the case of N41H, which moves more smoothly, the change might more prominently affect the biochemical properties of the filament with less effect on actin binding protein interaction, thus allowing for mutant filaments to be recycled slightly faster than in the PbS4Oc mutant.

### The role of divergent subdomain 3 amino acid residues

Combined mutation of subdomain 3 amino acid residues resulted in a chromobody localization that suggests these mutations could have a negative effect on filament stability or specific location. This was unexpected, since it is generally anticipated that mutations to canonical equivalents would render mutants that can form more stable filaments, as observed with N41H and PbS4Oc. It has been observed, using a similar chimera and mutagenesis approach in yeast, that conversion of yeast actin to mammalian alpha-actin equivalents resulted in

biochemical parameters that were not found between the two species [69]. This emphasizes that changes to actin dynamics or localization cannot always be correctly predicted and underlines the need to use appropriate probes and assays to measure such effects. The ATET/SEPQ mutant had a pronounced effect on parasite motility at multiple stages. Such striking phenotypic consequences further highlight the need for the parasite to have a sufficient pool of correctly localized filaments to ensure rapid motility and that the changes in *Plasmodium* actin sequence are not simply to reduce *Plasmodium* actin into a form that is incapable of polymerisation. As is the case in all cell systems, an optimal balance of actin filament assembly and disassembly is essential for all the processes in which actin is involved. The effects of the ATET/SEPQ mutations on actin kinetics should be further explored using *in vitro* biochemical methods. It would also be of interest to biochemically explore a combination mutant of both N41H and ATET/SEPQ to assess which of the residue changes has the dominant effect on actin dynamics.

Apart from very slight decreases in motile salivary gland sporozoites for E308P, individual subdomain 3 actin mutant lines did not have striking motility defects yet were defective in salivary gland invasion, suggesting that some modulation to actin dynamics is still occurring at the cellular level. Residue A272 is found directly on the H-plug, and T277 at the alpha-helical base of the plug [38]. The H-plug is located within the F-actin structure and electrostatically interacts with two neighbouring subunits, including the D-loop of the neighbouring monomer [61,62]. As with the D-loop, flexibility of the H-plug is important for filament formation. Emphasizing the importance of this region, mutations of the H-plug in different human alpha-actin isotypes are associated with various myopathies and aortic dysfunction [70]. The mutation of H-plug residues of *Plasmodium* actin has been studied *in vitro* [33]. Mutations K270M (which we had previously been unable to introduce) and A272W (A272S in this study) resulted in pronounced effects on phosphate release and significantly increased proportion of longer filaments. A high-resolution structure of the A272W mutant indicated that this mutation had an effect on the conformation of the A-loop of the neighbouring monomer, with the conformation of this loop providing critical interactions that determine filament stability and length [33]. It is reasonable to suggest that a similar effect could be occurring in A272S. Mutation of the position equivalent residue of 277 in yeast actin from an acidic residue to an apicomplexan residue [71] (mutant D275R) was detrimental for yeast cell growth *in vivo* without affecting polymerization properties *in vitro*. Interestingly, the mutation rendered the filaments more susceptible to cofilin action [72]. Mutation of *Plasmodium* actin T277 to an acidic residue could thus negatively affect ADF/cofilin function in severing filaments to an extent that influences the critical turnover specifically needed for salivary gland invasion.

E308 and T315 are located on the external side of the filament and are thus probably contributing indirectly to actin dynamics by mediating direct binding to actin binding proteins. Both residue positions have been implicated in myosin proximity in earlier studies [73,74], although these positions do not seem to feature prominently for interactions in the high resolution structures of both mammalian and *Plasmodium* actomyosin complexes [68,75,76]. These residue positions are in close proximity to tropomyosin in opisthokont systems [77,78]. Interestingly, tropomyosin-related proteins are conspicuously absent in the *Plasmodium* genome [26]. We anticipate that these divergent residues could be providing unique interactions for as-yet unidentified parasite specific actin binding proteins. Indeed, the failure of low concentrations of Jasplakinolide to rescue hemolymph sporozoite motility of the ATET/SEPQ mutant hints at an effect of these mutations beyond simply affecting filament stability, but rather points to a possible inability of specific actin binding proteins to bind this mutant and mediate its highly rapid dynamics. Testing of uninvestigated individual mutants in terms of their polymerisation properties and actin binding protein profiles will be important to tease out the distinct contributions of each of these residues and awaits further study.

## Conclusions

In this work, we have been able to delineate the contribution of different amino acid residues and their contribution to divergent actin function and localization. We identified that a single residue change in subdomain 2 (N41H) is sufficient to affect filament localization and strongly affects motility in hemolymph-derived sporozoites. We also identified that four amino acid residues have important roles in maintaining some form of stable yet dynamic filaments, with negative consequences for the parasite motility at two different stages. Modulating the actin sequence had serious adverse effects specifically in salivary gland invasion, suggesting optimal actin dynamics are required for this essential step in the parasite life cycle (**Fig 7**). Finally, we report the first visualization of actin filaments in the highly motile sporozoite using the actin chromobody. This revealed discrete locations of actin filaments in the parasite that were modified when actin dynamics were modulated by chemicals or mutagenesis. These approaches together have indicated that different amino acid residue substitutions have different effects, and therefore roles, in *Plasmodium* actin function and yet, irrespective of these differences, affect salivary gland penetration and ultimately transmission from the mosquito vector.

## Materials and methods

### Ethics statement

All animal experiments were performed according to the Federation of European Laboratory Animal Science Associations (FELASA) and Gesellschaft für Versuchstierkunde/Society for Laboratory Animal Science (GV-SOLAS) standard guidelines. Approval was granted by the responsible German authorities (Ethics Committee: Regierungsprasidium Karlsruhe). For all experiments, female four to six week old Swiss CD1 (for parasite propagation, blood stage growth rate assays, ookinete assays and mosquito feeds) and C57Bl/6 (for transmission studies) mice were used (obtained from Janvier Laboratories). *Anopheles stephensi* mosquitoes were reared and maintained by standard breeding methods.

### Generation of actin replacement mutant parasite lines

*Actin 1* ORF replacement constructs were generated as described before [36]. Briefly, a pair of complementary primers of approximately 25 to 30 base pairs containing the mutation(s) of interest were designed. These mutagenesis primers (primers 1–10, see **S1 Table**) were utilized to amplify the sequencing plasmid containing the codon modified actin 1 ORF. After confirming that these amplicons were of the correct sizes via agarose gel electrophoresis, 1.5 μL of DpnI restriction endonuclease was directly added (New England Biolabs, #R0176S (20 U/μL)) into the remaining 15 μL PCR product. This reaction mix was incubated at 37˚C/ 2 h to digest the parental methylated template. Next, 5 μL of this DpnI-digested PCR product was transformed into competent *E. coli* and the mutation verified via both restriction enzyme digestion and sequencing. The mutated ORFs were subsequently cloned into transfection vector Pb238 via BamHI and XbaI restriction sites. The transfection construct was linearised with SalI and PmlI, transfected into the actin recipient line and integration selected by negative selection using 5-fluorocytosine (1 mg/ml in drinking water) [36,79]. This rendered a line free of the selection cassette and contained the desired change in actin sequence. Isogenic parasites from the transfection mixture of integrants and resistant recipient line was obtained by limiting dilution. Asexual growth rate was determined in mice infected with single parasites during generation of clonal parasite lines in limiting dilution experiments. After the parasite clones were confirmed positive via PCR genotyping (**S1 Table**), Giemsa-stained blood smears of

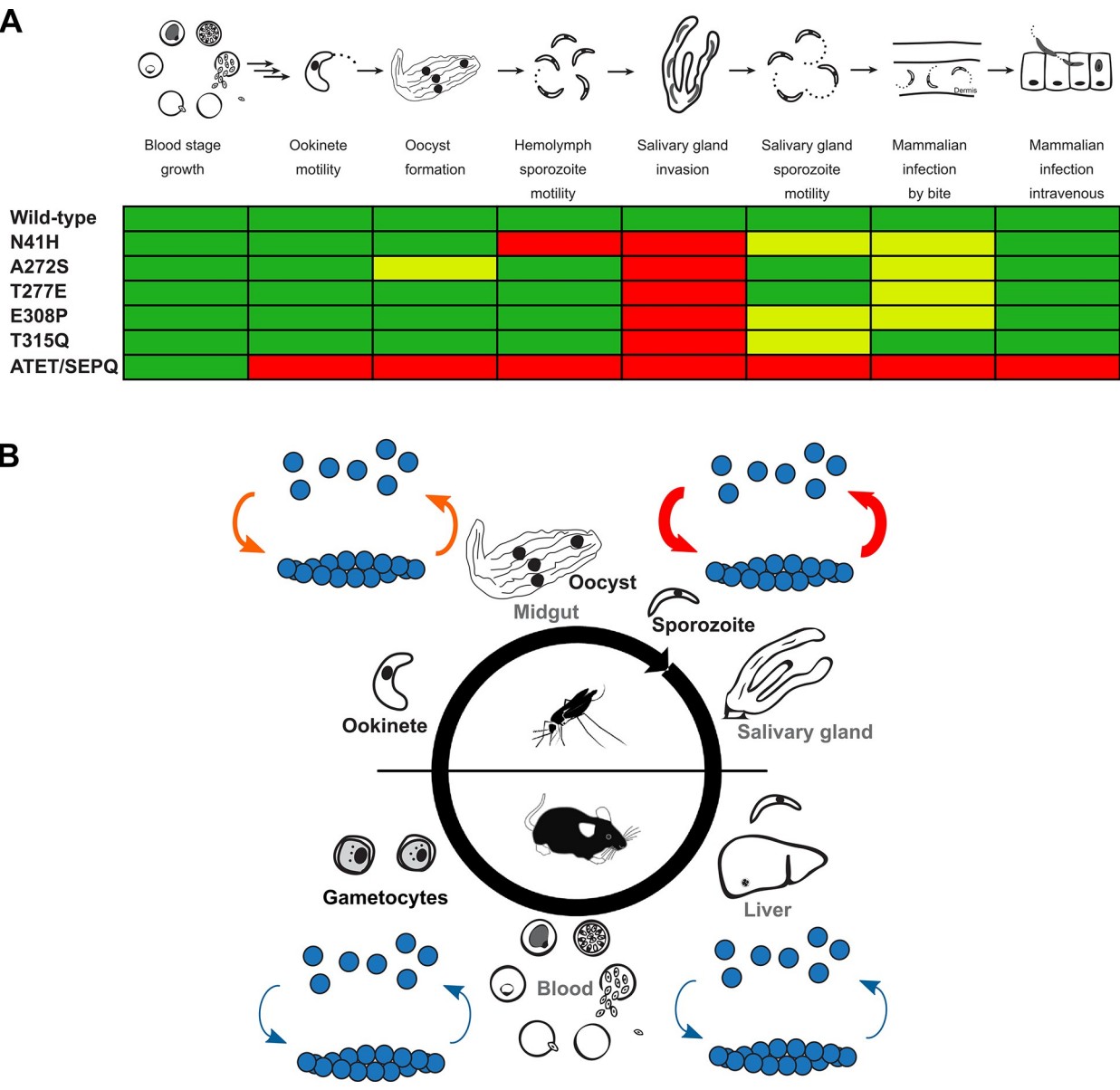

**Fig 7. Schematic representations of parasite susceptibility to a shift in actin dynamics across the *Plasmodium* life cycle.** (A) A linear representation of the parasite life cycle and a summary of the consequences of actin mutation. Green indicates no striking effect at that stage, while yellow and red colours indicates indicate a moderate or severe phenotype respectively. Ookinete motility for single mutants is presumed unaffected given the minor or undetectable differences in oocyst levels. (B) Our data suggest that parasites are less susceptible to changes in their actin dynamics in the mammalian host, with increasing susceptibility in the mosquito, particularly in sporozoites. Arrows of different thicknesses indicate increasing sensitivity to changes in actin dynamics, and very likely increasing dynamics, for these stages.

these clonal parasite lines day 8 post-infection (with a parasitemia of less than 1%) were used to back-calculate their asexual growth rate as described elsewhere [80].

## Generation of actin chromobody lines

Chromobody-emerald was amplified from plasmid [44] using primers 11 and 12 (**S1 Table**) to generate amplicon with BamHI and NotI flanking restriction sites. The amplified ORF was subsequently cloned into modified transfection vector Pb238 [36] via BamHI and NotI

restriction sites to produce a plasmid containing chromobody-emerald with the actin 1 promoter, 3' *dhfs* UTR, human DHFR selection cassette and homology arms for integration into chromosome 12 [22]. The construct was linearized with PvuI, transfected as above into the different actin lines, integration selected for with pyrimethamine, confirmed by genotyping (S1 Table) and clonal lines obtained by limiting dilution.

### *In vitro* ookinete culture and analysis

To develop ookinetes directly from whole blood, a donor mouse was infected by intraperitoneal injection of a frozen parasite cryostock. Parasites were allowed to grow in the infected mouse until a parasitemia of 1.5% to 2%. The parasites were harvested from the infected mouse via cardiac puncture and was used for fresh blood transfer of $20x10^6$ blood stage parasites intraperitoneally into naïve recipient mouse. Three days post-transfer, exflagellation centres were assessed as below (see mosquito infection). When more than one exflagellation event per field was observed the parasites were harvested from the recipient mouse via cardiac puncture and immediately transferred to 12 mL pre-incubated ookinete medium ((RPMI-1640, 25 mM Hepes, 300 mg l$^{-1}$ L-glutamine, 10 mg l$^{-1}$ hypoxanthine, 50 000 units l$^{-1}$ penicillin, 50 mg l$^{-1}$ streptomycin, 2 g l$^{-1}$ NaHCO$_3$, 20.48 mg l$^{-1}$ xanthurenic acid, 20% foetal bovine serum, pH 7.8) at 19˚C. The parasites were cultured without shaking at 19˚C for 21 h to allow ookinete development. Following 21 h of incubation, 1 mL aliquot of the ookinete culture was taken and pelleted at 7 000 rpm for 2 minutes at room temperature. The supernatant was removed and 3 μL of the pellet was used to make a Giemsa-stained smear to check for ookinete development.

To set up ookinete motility assay, Nycodenz purification was performed when high amount of ookinetes were observed in the Giemsa-stained smears while ookinete enrichment via brief centrifugation was performed when low amount of ookinetes were observed. To purify ookinetes using Nycodenz density gradient purification, 10 mL of the ookinete culture was carefully underlaid with 10 mL of 63% Nycodenz cushion in a 50 mL tube and spun at 1 000 rpm for 25 min at room temperature (no brake). After centrifugation, the "ring" of ookinetes formed at the interphase was harvested using a Pasteur pipette into a 15 mL tube. The purified ookinetes were pelleted at 1 000 rpm for 8 minutes at room temperature. The supernatant was removed and the pellet was resuspended with 1 mL of ookinete medium. To enrich ookinetes via brief centrifugation, 1 mL aliquot of the ookinete culture was pelleted at 7 000 rpm for 2 minutes at room temperature.

For imaging motile ookinetes, either pelleted 200 μL aliquot from Nycodenz purified ookinete or pelleted 1 mL aliquot directly from ookinete culture was resuspended in 20 μL of ookinete medium. Approximately 2 μL of these sample was mounted on a microscope slide, covered with a cover slip and sealed with paraffin. Ookinete motility was visualized using a wide-field microscope (Carl Zeiss, Axiovert 200M) with 63x oil objective. Images were acquired every 20 s for 15 min in the DIC channel (80 ms exposure). Movers were defined as those ookinetes that moved more than one parasite length. Average speed of ookinetes were calculated using the Fiji Manual Tracking plug-in.

### Mosquito infection

To feed a mosquito cage, a mouse was infected by intraperitoneal injection of a frozen parasite cryostock. Four days post-infection, the infected donor mouse was bled via cardiac puncture and $20x10^6$ blood stage parasites transferred into two naïve recipient mice through intraperitoneal injection. Three days post-transfer, exflagellation was assessed by incubation of a drop of tail blood for 10 minutes at 20˚C. Exflagellation centres were observed by light microscopy

(Carl Zeiss GmBH, Jena, Germany). In the event that >1 centre/field of view was observed, the mosquito infection procedure was continued. Prior to the cage feed, mosquitoes were starved for ≥4 hours by removing the salt and sugar pads. The two recipient mice were anaesthesized via intraperitoneal administration of ketamine-xylazine mixture (87.5 mg/kg ketamine, 12.5 mg/kg xylazine). The mice were placed on the cage first dorsally and then ventrally for~15 min each side to allow the mosquitoes to take blood meal for a total of 30 to 40 minutes. At the end of the mosquito infection, the mice were euthanized by cervical dislocation. The infected mosquito cage was incubated at 21˚C at 80% humidity, fed with 10% (v/v) saccharose with 0.05% PABA and 1% (v/v) NaCl. Mosquitoes used for infection are optimally between the age of 3 to 7 day-old.

### Characterisation of mosquito stage parasites

**Midgut oocysts analysis.**   To determine the infection rate of mosquitoes and the number of oocysts per infected mosquito, mercurochrome staining of mosquito midguts was performed. The midguts of 20 to 30 mosquitoes were harvested into a 1.5 mL tube containing 100 μL of PBS on ice 11 to 12 days after an infectious blood meal. 100 μL of freshly prepared 2% (v/v) NP-40 (in PBS) (Applichem, #A1694, 0250) was added to achieve a final concentration of 1%, to allow permeabilization of the dissected midguts at room temperature for 20 minutes. After 20 minutes, NP-40 solution was discarded and the permeabilized midguts were stained with 1 mL of 0.1% (w/v) mercurochrome solution (in PBS) for at least 30 min (or up to 2 h) at RT. Once the staining was completed, the stained midguts were washed with PBS until the supernatant was clear. The midguts were allowed to settle by gravity for 1 min to remove supernatant in between washing steps. After the last washing step, some supernatant was left on top of the settled midguts and they were transferred onto the microscope slide using a Pasteur pipette. These stained midguts were arranged using a needle and were gently covered with a coverslip. The stained midguts were visualized using a wide-field microscope (Carl Zeiss, Axiovert 200M) at 10x magnification (NA 0.5) in DIC channel with a green filter (38 HE Green Fluorescent Prot) and counted.

**Sporozoite extraction.**   Depending on the purpose of the experiments, sporozoites were harvested from infected mosquitoes between 14 to 24 days post-infection. Female mosquitoes were collected from the cage into a pre-chilled 15 mL tube. The tube was placed on ice to immobilize the mosquitoes. Sporozoites were typically harvested into RPMI-1640 medium (supplemented with 50 000 units.$l^{-1}$ penicillin and 50 mg.$l^{-1}$ streptomycin) for motility assays and counting or into PBS for intravenous injections. For isolation of midgut and salivary gland sporozoites, the immobilized mosquitoes were briefly placed into a Petri dish containing 70% ethanol, quickly dried on a paper towel and transferred into a second Petri dish containing PBS using a forceps. Midguts and salivary glands were harvested in PBS on a microscope slide under a dissecting microscope (Nikon). The harvested midguts and salivary glands were crushed using a pestle for 1 minute to crush the tissue and thereby release the sporozoites. For isolation of hemolymph sporozoites, mosquitoes collected into the pre-chilled 15 mL tube were incubated on ice for ~30 minutes to immobilize the mosquitoes for the harvesting process. A long-drawn Pasteur pipette (pre-filled with RPMI or PBS) was inserted into the thorax at a 90˚ angle and injected into the mosquito to flush out the hemolymph from a pre-cut abdominal incision. The hemolymph sporozoites were flushed out from the cut abdomen onto a parafilm and collected into a 1.5 mL tube on ice.

**In vitro sporozoite motility assay.**   For hemolymph sporozoites, hemolymph from 10 to 20 mosquitoes (between 14 to 16 days post-infection) were collected into a 1.5 mL tube on ice. The isolated hemolymph sporozoites were pelleted at 13 000 rpm/ 3 min/ RT. For salivary

gland sporozoites, 10 to 30 salivary glands (between 17 to 24 days post-infection) were harvested into a 1.5 mL tube containing 50 μL of RPMI on ice. The harvested salivary glands were crushed using a pestle for 1 minute to release the sporozoites. The sporozoite containing suspension was topped up to a final volume of 1 mL with RPMI-1640 medium (supplemented with 50 000 units.l⁻¹ penicillin and 50 mg.l⁻¹ streptomycin) and transferred to a 15 mL tube. To purify the sporozoites using density gradient purification [81], 3 mL of 17% (w/v) Accudenz (Accurate chemical, #AN7050) was carefully underlaid using a Pasteur pipette and the tube was centrifuged at 2 800 rpm for 20 minutes at room temperature (no brake). Using a pipette, the interphase (containing sporozoites) was collected (200 μL x 7 times) into a 1.5 mL tube, mixed by inverting the tube several times and spun down at 13 000 rpm for 3 min at room temperature. The supernatant was carefully discarded, leaving behind 15 to 20 μL of liquid as the pellet is not visible. To activate sporozoite motility, the sporozoite-containing pellet (either hemolymph-derived or salivary gland-derived) was resuspended in 100 μL of 3% (w/v) BSA (in RPMI-1640 medium; spun down at 13 000 rpm for 3 min at room temperature prior to use). The activated sporozoite solution was transferred to a 96-well optical-bottom plate (NUNC) and briefly centrifuged at 1 000 rpm for 3 minutes at room temperature to allow attachment of sporozoites at the bottom of the well. Sporozoite motility was visualized using a wide-field fluorescence microscope (Carl Zeiss, Axiovert 200M) at 63x magnification. Images were acquired every 1.5 s for 100 cycles at DIC channel (100 ms exposure) within 1 hour of BSA activation. Motility behaviors of the imaged sporozoites were categorized into 4 categories. 1) Movers were those that moved a minimum of 50 frames in either direction with no more than 10 frames of pause in between. 2) Partial movers were those that moved more than one sporozoite length but less than 50 frames. 3) Non-movers were those that did not move more than one sporozoite length including patch gliders, attached sporozoites and wavers. 4) Floaters were those that did not attach during the acquisition time. The average speed of movers over 100 frames was calculated using the Fiji Manual Tracking plug-in.

In order to assess whether Jasplakinolide can rescue the motility defects of the ATET/SEPQ mutant, hemolymph sporozoite extraction and preparation was performed as above. Several movies without compound were carried out before the addition of 1 μl of 100x concentration (5 or 10 μM) Jasplakinolide (final concentrations: 50 nM or 100 nM). The solution was mixed by gentle pipetting, the 96-well plate briefly recentrifuged at 1 000 rpm for 3 minutes and subsequently imaged for the remaining time in the 1 hour BSA activation timeframe. Motility behaviors were classed as above.

**Visualization of sporozoite actin dynamics.** To visualize actin dynamics in chromobody expressing hemolymph-derived and salivary gland-derived sporozoites, sporozoite motility was similarly activated with 100 μL of 3% (w/v) BSA (in RPMI-1640 medium) further supplemented with 1 μg/mL Hoechst. Images were acquired using wide-field microscopy as above in the GFP channel ($\lambda_{ex}$ 450 nm, $\lambda_{em}$ 515 nm, 100 ms exposure) for the chromobody, Hoechst channel ($\lambda_{ex}$ 350 nm, $\lambda_{em}$ 455 nm, 60 ms exposure) for Hoechst-stained nuclei and DIC (100 ms exposure) every 3 s for 25 cycles. To determine the effects of actin modulators on sporozoite actin dynamics, 1 μM of Cytochalasin D or Jasplakinolide was added into the untreated sporozoites, mixed and spun again at 1 000 rpm for 3 min at room temperature to allow sporozoite attachment. Localization of chromobody signals (untreated and treated) were imaged within 1.5 hours after BSA activation. Estimations of chromobody fluorescence intensity were performed by quantifying intensity of both the front and back ends of Jasplakinolide treated hemolymph sporozoites and subtracting nearby non-parasite background levels using a 0.639 μm² quantification circle. Background subtracted values were normalized to the average value obtained for the wild-type control.

Parasite transmission studies back into naïve mice were performed as described previously [36,80]. Given the reduced numbers of salivary gland resident parasites in mutant lines, 1 000 sporozoites were injected i.v.

## Supporting information

**S1 Fig. Generation and genotyping of *Plasmodium berghei actin 1* mutants.** (A) Integration schemes and the arrangement of the genomic locus in both untransfected (recipient line, RL, below) and transfected lines. The numbers indicate the primer combination used (see **S1 Table**) and the relative positions of those primers. (B) Representative gels of selected geno-typed parasite lines.
(TIF)

**S2 Fig. Jasplakinolide (Jas) treatment does not rescue the ATET/SEPQ motility phenotype.** Hemolymph sporozoites were extracted and activated as described in the Material and Methods section. Low concentrations of Jas were added after several movies (without Jas) were acquired. Addition of Jas did not improve motility.
(TIF)

**S3 Fig. Representative montages of sporozoites classed as non-movers.**
(TIF)

**S4 Fig. Generation and genotyping of the *Plasmodium berghei actin chromobody-emerald* line.** (A) Integration schemes and the arrangement of the genomic locus in both untransfected and transfected lines. The numbers indicate the primer combination used (see **S1 Table**) and the relative positions of those primers. (B) Agarose gel electrophoresis of generated isogenic lines.
(TIF)

**S5 Fig. Characterization of the actin chromobody line.** (A) Schematic representation of the *Plasmodium* life cycle with late mosquito stages indicated with a dashed line. (B) Hemolymph sporozoite motility patterns were analysed using a 2D motility assay and revealed no signifi-cant differences to a parallel control background cage. Bars represented as mean ± standard deviation for repeats between at least two technical repeats from a single infection each. Fischer's exact test. (C) Speeds of moving hemolymph sporozoites indicate that chromobody sporozoites might move slightly faster, yet were in comparable ranges and likely represent experimental variations despite being significantly different. Red line indicates median value. Mann-Whitney test, ****p<0.0001. (D) The actin chromobody line displays a moderate decrease in salivary gland invasion. Red line indicates median value. Dots represent technical repeat counts from day 18 and day 19 post-infection (one mosquito infection for the control line and two independent infections for the actin chromobody line). (E) Salivary gland sporo-zoites are similarly motile between control and chromobody lines. Bars represented as mean ± standard deviation for repeats between at least two technical repeats. Fischer's exact test. (F) Actin chromobody expressing salivary gland sporozoites showed similar range of speeds when compared to the wild-type control but were significantly slower on average. Val-ues are in the normal range for wild-type parasites and likely represent experimental variations despite being significantly different. Red line indicates median value. Mann-Whitney test, ****p<0.0001.
(TIF)

**S6 Fig. An example of a motile sporozoite with a change in signal distribution.** The white arrowhead indicates the rear of the sporozoite. In this case, a noticeable signal develops at the

back with time. Scale bar: 5 μm. Changes in signal distribution were observed infrequently.
(TIF)

**S7 Fig. Coarse-grained protein level estimations of mutants analysed with the actin chromobody. (**A) Individual readings from separate replicates (individual imaging sessions) for each parasite line. Each data point reflects a background subtracted front and back sporozoite intensity normalized to the average wild-type control. Note that N41H median intensities were variable between duplicate imaging sessions. (B) Data of replicates in (A) combined. Red line indicates median. Mutants generally did not have major changes in their normalized fluorescence levels, with the exception of N41H which might have reduced functional protein levels. Mann-Whitney test, **p = 0.001 ****p<0.0001. ATET/SEPQ had a minor increase in intensity that likely reflects assay limitations and experimental variation despite being statistically significant.
(TIF)

**S1 Data. Numerical data used to generate plots in figures.**
(XLSX)

**S1 Table. Primers used in this study.**
(DOCX)

## Acknowledgments

We thank Miriam Reinig for mosquito rearing and Julia Sattler for technical assistance. We further thank Julia Sattler, Markus Ganter, Franziska Hentzschel and Markus Meissner for helpful discussions during the course of this study. We especially thank Markus Meissner for the kind gift of the chromobody plasmid.

## Author Contributions

**Conceptualization:** Friedrich Frischknecht, Ross G. Douglas.

**Formal analysis:** Michelle Yee, Ross G. Douglas.

**Funding acquisition:** Friedrich Frischknecht.

**Investigation:** Michelle Yee, Tobias Walther, Ross G. Douglas.

**Methodology:** Tobias Walther, Ross G. Douglas.

**Project administration:** Friedrich Frischknecht.

**Supervision:** Friedrich Frischknecht, Ross G. Douglas.

**Visualization:** Michelle Yee, Ross G. Douglas.

**Writing – original draft:** Friedrich Frischknecht, Ross G. Douglas.

**Writing – review & editing:** Michelle Yee, Friedrich Frischknecht, Ross G. Douglas.

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
