## [Decision Letter · Decision Letter 0]

21 Mar 2022

Dear Dr Douglas,

Thank you very much for submitting your manuscript "Divergent Plasmodium actin residues are essential for filament localization, mosquito salivary gland invasion and malaria transmission" for consideration at PLOS Pathogens. As with all papers reviewed by the journal, your manuscript was reviewed by members of the editorial board and by several independent reviewers. In light of the reviews (below this email), we would like to invite the resubmission of a significantly-revised version that takes into account the reviewers' comments.

As you can see from the reviewer comments below, all three reviewers are supportive of your work. However, all of them raise some concerns, which you should address. In particular, reviewers 1 and 3 ask for important control experiments to be included. In addition to the reviewers' comments, it would be helpful for the readers to include a more detailed figure showing the contacts made by the residues that have been mutated in this study in the actin filament. In figure 1A, it would be better to have colors that are easier to distinguish from each other.

We cannot make any decision about publication until we have seen the revised manuscript and your response to the reviewers' comments. Your revised manuscript is also likely to be sent to reviewers for further evaluation.

Sincerely,

Inari Kursula, Ph.D.

Guest Editor

PLOS Pathogens

James Kazura

Section Editor

PLOS Pathogens

Kasturi Haldar

Editor-in-Chief

PLOS Pathogens

orcid.org/0000-0001-5065-158X

Michael Malim

Editor-in-Chief

PLOS Pathogens

orcid.org/0000-0002-7699-2064

As you can see from the reviewer comments below, all three reviewers are supportive of your work. However, all of them raise some concerns, which you should address. In particular, reviewers 1 and 3 ask for important control experiments to be included. In addition to the reviewers' comments, it would be helpful for the readers to include a more detailed figure showing the contacts made by the residues that have been mutated in this study in the actin filament. In figure 1A, it would be better to have colors that are easier to distinguish from each other.

Reviewer's Responses to Questions

**Part I - Summary**

Reviewer #1: Yee et al explore the functional role of divergent Plasmodium actin residues. Previous work showed that subdomain 1 and 4 of Plasmodium actin contributed to motility in mosquito stage parasites. Subdomain 2 and 3 proved essential because these domains could not be exchanged with the mammalian sequence. Here the sequences of subdomains 2 and 3 were compared with mammalian sequences and specific conserved residues of interest were identified based on three well justified criteria. Two residues in domain two were mutated to the conserved mammalian residue, but only N41H could be recovered. Five residues in subdomain three were mutated, but only four were recovered. A272S, T277E, E308P, and T315Q. In addition to the single point mutants a quadruple subdomain 3 mutant ATET/SEPQ was generated. The authors then infected mosquitos with mutant gametocytes and assessed oocyst levels in the mosquito midgut. A272S had a modest reduction in oocyst levels, but the quadruple mutant was severely diminished indicating that subdomain 4 plays an important role in mosquito specific actin dynamics. This interpretation is supported by dramatically reduced ookinete motility. Next the authors examined the ability of sporozoites to establish infection in the salivary gland. Compared to the control all the point mutants had reduced salivary gland occupancy. Subdomain 2 N41H mutants isolated from hemolymph and salivary glands was found to have reduced motility. The subdomain 3 E308P mutant was also found to have reduced motility but the quadruple mutant could not be examined because only two sporozoites were observed. The authors tested the ability of the mutants to be transmitted. Natural transmission was found to be lower as expected due to reduced salivary gland parasite loads, but i.v. mouse infections proceeded similar to the controls indicating that the substitutions found in Plasmodium actin are likely present to optimize mosquito stage infection. To gain further insight into how the point mutants impact actin dynamics the authors utilized an actin chromobody to visualize actin. They found an asymmetric localization of actin on one side of mobile sporozoites that could be disrupted with either CytoD (even distribution of the chromobody) or Jas (localization to both ends). These changes in distribution allow for interpreting the impact of the substitutions in terms of stabilizing or destabilizing actin filaments. Introduction of the actin chromobody into N41H led to chromobody accumulation at both ends of the cell indicating that this mutation stabilizes actin. The ATET/SEPQ mutant resembled CytoD treated cells indicating that the mutations led to destabilized actin perhaps due to loss of actin binding protein interactions. The manuscript is well organized and well written. The work convincingly demonstrates that finely tuned actin dynamics are essential for efficient organ entry into the mosquito and ultimately for transmission.

Reviewer #2: The overall goal of this study is to assess the effect of several individual (or a group of 4) point mutants in Plasmodium actin 1 at four different stages of the parasite lifecycle, encompassing both the mosquito and the rodent hosts. Actin is generally highly conserved, but Plasmodium actin 1 has regions that are quite divergent. Mutation of a subset of these divergent residues in subdomains 2 and 3 to the more commonly found amino acid at that positon resulted in viable parasites that were selected for further phenotypic investigation.

Interestingly, some of the major effects of the mutations were seen in transmission to and through the insect, and in particular resulted in impairment in salivary gland invasion, suggesting that this step is potentially a difficult barrier that the parasite must overcome to complete its lifecycle. A novel feature of this study is the use of the actin-chromobody for visualization of actin in sporozoites (Plasmodium actin has been notoriously difficult to visualize until quite recently).

I find the study thoughtfully done and interesting. It is both a logical follow-up of their earlier studies with actin subdomain swaps, and also provides a roadmap for future investigation of several of these mutations with regard to how they might affect interaction with actin-binding proteins and thus alter filament dynamics.

Reviewer #3: This work reports in vivo effects of actin mutations in Plasmodium at different life stages. Previous studies have shown that actin is essential for Plasmodium and only lethal mutants had been characterized. The authors focused on actin residues that are different from mammalian actins and found several point mutations that keep the parasite viable but cause defects in their behaviors. Their analysis suggests different requirements of actin filament dynamics at each life stage. They also utilized actin chromobody to visualize actin localization in motile sporozoite, which should become a very useful tool for future cytoskeletal research in Plasmodium. Overall, this research is well done with quantitative analysis and careful interpretation and should provide significant advances to the related field.

**Part II – Major Issues: Key Experiments Required for Acceptance**

Reviewer #1: Figure 5. Many actin probes have an impact on actin dynamics which the authors show has a role in Plasmodium transmission. A control experiment measuring the speed of movement such as in Fig 3 C and F for true WT sporozoites versus ones expressing the actin chromobody would reveal whether actin dynamics are appreciably altered by the chromobody. The use of probes is necessary in order to visualize how different conditions impact actin localization, so the aim would be to determine the level to which the chromobody impacts actin dynamics.

Figure 6. The authors only examined the ATET/SEQP mutant for subdomain 3. They found that together these mutations resulted in actin destabilization. We are left to wonder if each residue is contributing to destabilization or if there might be some residues that are more important as suggested by the single mutant analysis in Figures 1-4. Therefore, the authors should complete their analysis by examining chromobody localization in each of the single mutants of subdomain 3. This would allow some correlation with the speed of movement reported in Fig 3C and F.

Reviewer #2: none

Reviewer #3: 1. Point mutations can often affect the protein levels due to changes in protein stability in vivo. If the protein level is significantly altered, it can have an impact on the phenotype. Therefore, the authors should confirm that the actin protein levels were not altered in the mutants that they analyzed. It is not clear whether an actin antibody is available to detect Plasmodium actin on Western blot. Even if it is not available, actin should be a major 42 kDa protein and could be estimated from an SDS-PAGE gel.

**Part III – Minor Issues: Editorial and Data Presentation Modifications**

Reviewer #1: Line 1018 remove “infections”

Mammalian actin is likely optimized for function at 37C. Mosquitos are grown at cooler temperatures. It is known that actin point mutations can alter sensitivity to temperature. It seems that Plasmodium actin has to be optimized for both conditions, but this was not discussed. It seems that all the motility studies were performed at room temperature. If the authors had access to a temperature controlled stage it would be interesting to test if motility of the mutants can be rescued with higher temperatures. This would help explain why infectivity defects are present in mosquitos but not in mice.

Jas stabilizes actin filaments, it would be interesting to test if low doses of Jas could rescue the speed of the ATET/SEPQ quadruple mutant ookinetes. It is a simple experiment that if it worked would be informative. If it failed it might indicate that the defect is due to the inability of specific ABPs to bind to the quadruple mutant.

Not a proposed experiment for this study, but in addition to changes in temperature optimization there could be different actin binding proteins functioning in mosquito stages. An interesting future experiment would be to use the chromobody in proximity labeling experiments. TurboID or another biotin ligase fused to the chromobody could be used to identify which proteins associate with actin in different Plasmodium stages. This same experiment in the ATET/SEPQ mutant would also directly test the idea that specific ABPs are no longer associating with actin filaments in the mosquito stages.

Reviewer #2: Minor:

1. Figs. 2B, 2F, 3C, 3F make the individual data points smaller so they don’t merge.

2. In Fig. 3B don’t break the Y-axis (doesn’t save that much space with the breaks you made and looks strange).

Reviewer #3: 1. The actin localization in Figs. 5 and 6 are somewhat difficult to interpret. Without DIC/Nomarski images, the outlines of the parasites cannot be seen. For example, when actin localizes to both ends, is actin localized to the very ends or somewhere near the tips? Some of the representative images should be shown as overlays between fluorescent and DIC images to clearly demonstrate the locations of actin-rich regions.

2. It would be helpful for readers if a table for figure summarizing the results of mutations and phenotypes.

PLOS authors have the option to publish the peer review history of their article (what does this mean?). If published, this will include your full peer review and any attached files.

Reviewer #1: No

Reviewer #2: No

Reviewer #3: **Yes: **Shoichiro Ono
---

## [Decision Letter · Decision Letter 1]

29 Jul 2022

Dear Douglas,

We are pleased to inform you that your manuscript 'Divergent Plasmodium actin residues are essential for filament localization, mosquito salivary gland invasion and malaria transmission' has been provisionally accepted for publication in PLOS Pathogens.

Best regards,

Inari Kursula, Ph.D.

Guest Editor

PLOS Pathogens

James Kazura

Section Editor

PLOS Pathogens

Kasturi Haldar

Editor-in-Chief

PLOS Pathogens

orcid.org/0000-0001-5065-158X

Michael Malim

Editor-in-Chief

PLOS Pathogens

orcid.org/0000-0002-7699-2064

Reviewer Comments (if any, and for reference):

Reviewer's Responses to Questions

**Part I - Summary**

Reviewer #1: The authors have addressed all of my concerns with added text and experiments, I am satisfied with the revisions.

Reviewer #2: The overall goal of this study is to assess the effect of several individual (or

a group of 4) point mutants in Plasmodium actin 1 at four different stages of the parasite

lifecycle, encompassing both the mosquito and the rodent hosts. Actin is generally highly

conserved, but Plasmodium actin 1 has regions that are quite divergent. Mutation of a

subset of these divergent residues in subdomains 2 and 3 to the more commonly found

amino acid at that position resulted in viable parasites that were selected for further

phenotypic investigation. Interestingly, some of the major effects of the mutations were

seen in transmission to and through the insect, and in particular resulted in impairment in

salivary gland invasion, suggesting that this step is potentially a difficult barrier that the

parasite must overcome to complete its lifecycle. A novel feature of this study is the use of

the actin-chromobody for visualization of actin in sporozoites (Plasmodium actin has been

notoriously difficult to visualize until quite recently). I find the study thoughtfully done and

interesting. It is both a logical follow-up of their earlier studies with actin subdomain swaps,

and also provides a roadmap for future investigation of several of these mutations with

regard to how they might affect interaction with actin-binding proteins and thus alter

filament dynamics.

Reviewer #3: This work reports in vivo effects of actin mutations in Plasmodium at different life stages. The revised manuscript clarified all of my previous concerns.

**Part II – Major Issues: Key Experiments Required for Acceptance**

Reviewer #1: None.

Reviewer #2: none

Reviewer #3: None.

**Part III – Minor Issues: Editorial and Data Presentation Modifications**

Reviewer #1: None.

Reviewer #2: none

Reviewer #3: None.

PLOS authors have the option to publish the peer review history of their article (what does this mean?). If published, this will include your full peer review and any attached files.

Reviewer #1: No

Reviewer #2: No

Reviewer #3: **Yes: **Shoichiro Ono

---

## [Editor Report · Acceptance letter]

18 Aug 2022

Dear Douglas,

We are delighted to inform you that your manuscript, "Divergent *Plasmodium* actin residues are essential for filament localization, mosquito salivary gland invasion and malaria transmission," has been formally accepted for publication in PLOS Pathogens.

Best regards,

Kasturi Haldar

Editor-in-Chief

PLOS Pathogens

orcid.org/0000-0001-5065-158X

Michael Malim

Editor-in-Chief

PLOS Pathogens

orcid.org/0000-0002-7699-2064